# More than just Stem Cells: Functional Roles of the Transcription Factor Sox2 in Differentiated Glia and Neurons

**DOI:** 10.3390/ijms20184540

**Published:** 2019-09-13

**Authors:** Sara Mercurio, Linda Serra, Silvia K. Nicolis

**Affiliations:** 1Department of Biotechnology and Biosciences, University Milano-Bicocca, 20126 Milano, Italy; 2CNRS, Inserm, iBV, Université Côte d’Azur, 06108 Nice, France

**Keywords:** sox2, neurons, glia, transcription factors, neural stem cells, cerebellum, thalamus, dorsolateral geniculate nucleus, visual cortex, Bergmann glia, Müller glia

## Abstract

The Sox2 transcription factor, encoded by a gene conserved in animal evolution, has become widely known because of its functional relevance for stem cells. In the developing nervous system, Sox2 is active in neural stem cells, and important for their self-renewal; differentiation to neurons and glia normally involves Sox2 downregulation. Recent evidence, however, identified specific types of fully differentiated neurons and glia that retain high Sox2 expression, and critically require Sox2 function, as revealed by functional studies in mouse and in other animals. Sox2 was found to control fundamental aspects of the biology of these cells, such as the development of correct neuronal connectivity. Sox2 downstream target genes identified within these cell types provide molecular mechanisms for cell-type-specific Sox2 neuronal and glial functions. SOX2 mutations in humans lead to a spectrum of nervous system defects, involving vision, movement control, and cognition; the identification of neurons and glia requiring Sox2 function, and the investigation of Sox2 roles and molecular targets within them, represents a novel perspective for the understanding of the pathogenesis of these defects.

## 1. Introduction

### 1.1. Sox2 as a “Stem Cell Gene”

The transcription factor Sox2 has become widely known because of its functional connection to the stem cell state (reviewed in references [1,2,3,4]). In mammals, Sox2 is expressed, and required, from the earliest stages of embryonic development in the pluripotent stem cells of the blastocyst inner cell mass; its knock-out in mouse causes early embryonic lethality [5]. Conversely, Sox2 can reprogram differentiated cells to induced pluripotent stem cells (iPSCs), acting with a small number of other factors [6]. At later developmental stages, Sox2 is maintained (at least preferentially) in stem cells of different tissues [1], among which, prominently, the developing (and postnatal) nervous system [2,7]. Here, Sox2 is highly expressed in neural stem/precursor cells, constituting the ventricular zone of the developing neural tube and, in general, it is downregulated in differentiating cells as they progressively move out of the ventricular zone into the outer layers of the neural tube. Sox2 conditional deletion in the developing neural tube impacts neural stem cell (NSC) function, both in vitro and in vivo [2,8,9]. Recently, Sox2 was also found to be absolutely required for the development of the mouse olfactory neuroepithelium [9,10], a neurogenic epithelium containing Sox2-positive, long-term self-renewing stem-like cells [11].

### 1.2. SOX2 and Neurodevelopmental Genetic Disease

SOX2 heterozygous mutations in humans lead to a characteristic spectrum of nervous system abnormalities that includes eye defects and hippocampal defects, together with intellectual disability, seizures, and motor control defects [12,13,14]. In mouse models, severe eye defects are reproduced following Sox2 conditional deletion in the neural stem/progenitor cells of the developing retina [4,15]. Hippocampal defects are reproduced in mouse models of conditional Sox2 deletion in the neural tube at mid-embryogenesis (embryonic day 11.5, E11.5), and are explained by a loss of stem cells of the hippocampus dentate gyrus [8]. Administration of a drug, mimicking the effect of a cytokine (SHH) lost from the mutant hippocampus, encoded by a Sox2 target gene, rescues hippocampal NSC, and hippocampal growth [8]. Sox2 hippocampal function is likely to contribute to the seizures, as well as the cognitive defects, observed in patients.

Earlier deletion of Sox2 (E10.5) also compromises the development of the primordia of the basal ganglia in the ventral forebrain (medial ganglionic eminence, MGE), pointing to stage-specific functions of Sox2; in the context of the early MGE, an important downstream target of Sox2 is the gene encoding transcription factor Nkx2.1, a master regulator of ventral forebrain development [9].

Sox2 is very conserved in evolution, with homologs in Drosophila, C. elegans, chick, and zebrafish, in addition to mouse and human [3]. It is noteworthy that some Sox2 homologs in invertebrates can replace mouse Sox2 function in preserving embryonic stem cell fate [16].

### 1.3. Recent Perspectives on SOX2 Molecular Mechanisms of Action

SOX2 targets, and molecular modes of action, in NSC, are the object of strong interest (reviewed in references [2,3,17]). These issues have been recently investigated from a genomic perspective, using NSC cultured in vitro from the normal, or Sox2-deleted, mouse brain as a model system [18]. In brain-derived NSC, ChIPseq shows that SOX2 binds predominantly to distal enhancers, connected to promoters via RNApolII-mediated long-range interactions, identified by ChIA-PET analyses [18]. Distal SOX2-bound enhancers are active in vivo, in the developing forebrain of transgenic zebrafish and mouse embryos. Sox2-deleted cells show downregulation of some 700 genes by RNAseq; these genes are highly enriched in connections with SOX2-bound enhancers, and SOX2 loss from these interactions is the single most important factor underlying differences in gene expression in Sox2-mutant cells (more than SOX2 loss at promoters) [18]. Sox2 loss leads to the loss, or decrease, of a high number of long-range interactions, pointing to Sox2 as an “architectural” factor in 3D functional chromatin organization [18]. Of note, genes bound by SOX2 and involved in long-range interactions include many genes involved in neurodevelopmental diseases, showing phenotypic overlap with the abnormalities detected in Sox2-deficient patients, or Sox2-mutant mice. In addition, the functional study of genes regulated by Sox2 through long-range interactions identified Socs3, encoding a signaling molecule, as an important mediator of Sox2 function in maintaining NSC self-renewal [18]. Intriguingly, SOX2-bound genes also include genes that are still transcriptionally silent in NSC but will be activated following neuronal or glial differentiation (some of which will be discussed later in this Review). This suggests the possibility that SOX2-bound chromatin loops constitute a form of “priming” of neural genes for later activity in differentiation. SOX2 was also found to bind to genes carrying “bivalent” epigenetic histone modifications characteristic of “poised” genes, meaning genes that will become fully active only following differentiation. Conditional ablation [8] of SOX2 in adult hippocampal NSC, impaired the activation of pro-neural and neurogenic genes, during differentiation, supporting a functional role for SOX2 binding in NSC in establishing a permissive epigenetic state of neuronal/glial genes [19]. Recently, an additional architectural role of Sox2 in NSC chromatin was suggested by the finding of its interaction with the nucleoporin Nup153, a protein that is part of the nuclear pores [20]; SOX2 and Nup153 share a high fraction of binding sites on DNA, and their binding is enriched on genes deregulated following downregulation of Nup153. It is possible that Sox2 acts on chromatin architecture at the level of long-range interactions maintenance, as well as by possibly tethering genes to the nuclear pore. It will be interesting to ask if the SOX2-bound regions involved show any overlap.

The evidence summarized above demonstrates critical roles for Sox2 in NSC (recently reviewed more in detail in references [2,17]) and begins to unravel molecular mechanisms and targets mediating Sox2 function.

However, is Sox2 function in neural development limited to its roles in NSC?

Recent evidence shows that this is not the case. In this Review, we will present and discuss the unexpected critical functions played by Sox2 within specific classes of differentiated glial and neuronal cells.

## 2. Sox2 Functions in Differentiated Glia and Neurons

NSC produce, by controlled proliferation and differentiation, the various types of neuronal and glial cells that constitute the mature nervous system. Typically, NSC differentiation involves the downregulation of Sox2 expression. However, expression studies (e.g., by immunofluorescence with anti-SOX2 antibodies) revealed a high level of SOX2 within specific types of differentiated glial and neuronal cells; these studies, in turn, prompted functional investigations of Sox2 roles within these cells (Table 1).

### 2.1. Sox2 in Glial Cells

#### 2.1.1. Bergmann Glia and Müller Glia

##### Bergmann Glia in the cerebellum

During development, Sox2 is expressed in the primordium of the cerebellum, and, postnatally, it is maintained in the Bergmann glia (BG), a unipolar cell type (that originates from radial glia) with the soma in the Purkinje cell (PC) layer and their radial processes going through the molecular layer and reaching the pia [40] (Figure 1A). BG are essential for granule cell migration during development and in postnatal life, and they are important for Purkinje cells function by regulating ion homeostasis and synapse stability (reviewed in references [41,42]). Conditional knock-out of Sox2 in the developing cerebellum (mediated by Wnt1-Cre recombinase) results, postnatally, in hypoplasia of the central portion of the cerebellum, the vermis, and ataxia. In addition, a cell type that appears particularly affected by Sox2 loss is the Bergmann glia. BG cell bodies are found outside of the Purkinje cell layer, where they are usually localized, and their radial morphology is lost in Sox2 mutants (Figure 1A). Other glial cells that express Sox2, like parenchymal and prospective white matter astrocytes, appear unaffected by Sox2 loss, indicating a specific requirement for Sox2 in BG, but not in other glial cell types [29]. Since Wnt1-Cre-mediated Sox2 deletion occurs during early phases of embryogenesis, the observed phenotype could have been due to a problem occurring in the first phases of Bergmann glia differentiation from radial glia [40]. However, even postnatal conditional ablation of Sox2 in Bergmann glia—with a tamoxifen-inducible-Cre recombinase knocked-in the GLAST locus—showed that Sox2 is required to maintain BG radial morphology and their correct position in the PC-layer, in agreement with Sox2 maintaining an important role in postnatal, fully differentiated BG (Figure 1A). Interestingly, this later Sox2 ablation was accompanied by mild ataxia, while the vermis appeared normal, suggesting a likely direct role of Sox2-expressing BG in regulating neuronal activity [29]. One important role of BG that influences neuronal function, is the removal of glutamate from the synaptic cleft via the glutamate transporter GLAST. Defects in glutamate uptake in humans lead to ataxia [43]. This function of BG seems to be influenced by Sox2 loss since Sox2 mutant BG were found defective in glutamate uptake [29].

What genes does Sox2 regulate that mediate its function in determining the position, the morphology, and the activity of BG?

To date, no direct Sox2 targets in BG have been identified, but many Sox2 targets in NSC and glial cells in culture [18,29,44] are also expressed in BG and could mediate Sox2 function. In addition, mutations in some of these potential targets lead to ataxia and BG defects reminiscent of the ones due to Sox2 loss, for example mutations in the tumor suppressor adenomatous polyposis coli (APC) [45] and in the transcription factor Sox4 [46].

Finally, it has been reported that BG translocation is a marker for vanishing white matter disease, a severe leukoencephalopathy due to mutations in the eukaryotic translation initiation factor 2B (eI2B) and that glial abnormality may be important in the pathogenesis of the disease [47].

##### Müller Glia in the Retina

Another macroglia type that expresses Sox2 is the Müller glia in the retina [4,27,28]. Müller glia (MG) develop from retinal progenitor cells (RPC) after they have given rise to all the different retinal neurons [48]. MG have a radial morphology that spans across the whole retina; MG cells have their cell bodies in the inner nuclear layer (INL) (Figure 1B). MG are essential for the establishment and maintenance of the laminar organization of the postnatal retina, and for providing trophic support to neurons [49,50]. Ablation of Sox2 at the height of MG genesis in mouse (P5) using a tamoxifen inducible Cre recombinase line (GLAST-Cre-ER) led to the disruption of MG processes, that worsened with time and led to a thinner retina. In the Sox2 mutants, adherens junctions were shorter and not oriented correctly. The lack of Sox2 not only affected MG, but also the development of neuronal processes that probably required a functional MG to correctly differentiate [27].

In addition, MG cell bodies, usually localized in the INL, were located ectopically in the outer plexiform layer (OPL) and outer nuclear layer (ONL) in Sox2 mutants [27](Figure 1B). This same phenotype was also obtained by ablating Sox2 postnatally in differentiated MG with a tamoxifen inducible Cre recombinase (CAGG-Cre-ER); in this mouse model, retinas were treated with Tamoxifen in vitro, cultured for a few days and imaged with time-lapse videos. The time-lapse videos showed that the mis-localization of MG was an indication that they had re-entered the cell cycle and, therefore, interkinetic nuclear migration had occurred. MG display a similar nuclear movement when they re-enter the cell cycle following retinal damage [28]. In conclusion, Sox2 has a key role in maintaining MG shape and quiescence state, in order to prevent their premature re-entering of the cell cycle and subsequent depletion through cell divisions.

What genes does Sox2 regulate that mediate its function in determining the morphology, the position, and the quiescence of MG?

The Notch signaling pathway has been shown to be important, not only to regulate NPC fate decisions but also to maintain MG identity [51,52]. In addition, Notch1, a downstream target of Sox2 in RPC [15,53], is downregulated in Sox2 mutant MG, and ectopic activation of the Notch pathway in Sox2 mutant MG can rescue the radial morphology. However, Notch signaling does not seem to be involved in the maintenance of the MG quiescence state [28].

##### Can Sox2 Expressing BG and MG Include Cells with Stem Cell Features?

Both BG and MG express markers of stem cells, including Sox2, suggesting that they might have the capacity to re-enter the cell cycle and produce new neurons and glia upon damage. In the adult cerebellum, cells with properties of NSC-like cells were found to be localized in the Purkinje cell layer, intermingled with BG, and to express Sox2 [54]. MG maintains neurogenic capacity in the adult retina, and it has been shown to be able to re-enter the cell cycle in response to retinal injury [55,56]. These findings indicate that Sox2 may retain some of its stem cell-related functions, acting within subpopulations of differentiated glia, as they occasionally resume stem cell character.

#### 2.1.2. Myelinating Glial Cells: Oligodendrocytes and Schwann Cells

In vertebrates, axons are wrapped by myelin sheaths that allow rapid information transmission. Myelination is essential for the proper function of axons, and defects in its development or myelin damage can lead to serious diseases in humans like leukodystrophies and multiple sclerosis.

##### Oligodendrocytes

Oligodendrocytes are the myelinating glia of the central nervous system, and they originate from neural stem cells after neurons have been generated. Sox2 is expressed in oligodendrocyte progenitor cells (OPCs) during development and in early differentiating oligodendrocytes, while Sox2 expression in OPCs in the adult brain is still controversial [22,23,24,25]. Overexpression of Sox2 and Sox3 in oligodendrocytes precursors cells (OPC) converts them to neural stem cells [57], however, the role of Sox2 in differentiating oligodendrocytes was not clear until recently.

Conditional ablation in the oligodendrocyte lineage of Sox2 was generated in mice (by Sox10-Cre recombinase that leads to complete ablation by E15.5), and while no effect on proliferation, migration, and survival were observed at embryonic stages, at early postnatal stages terminal differentiation was impaired [22] (Figure 1C). When OPC proliferation was analyzed at postnatal stages, a strong reduction of OPCs was instead observed [24,25]. Since Sox2 expression peaks in the early stages of oligodendrocyte differentiation, its deletion, specifically at this stage of the oligodendrocyte lineage, was obtained (by CNP-Cre recombinase), and a diminished rate of oligodendrocyte generation was observed, accompanied, in the forebrain, by hypomyelination [23,24] (Figure 1C).

One way in which Sox2 is thought to influence myelination is by inhibiting the expression of miR145, a negative regulator of genes important for oligodendrocyte differentiation like Sox9, Med12, and Myrf [22] (Figure 1C). In addition, Sox2 binds the myelin basic protein promoter, but it is a much weaker activator than Sox10 (a marker of the whole oligodendrocyte lineage); this observation led to the idea that Sox2 could be important to poise the chromatin for Sox10, since SoxB1 factors have been shown to do this in neural stem cells [58,59].

##### Schwann Cells

Schwann cells (SC) originate from the neural crest and are the myelinating glia of the peripheral nervous system, involved in myelinating axons during normal development and in their remyelination following injury. While many positive regulators of myelination in the PNS have been identified, including Krox20 and Sox10, just a few negative regulators have been characterized, and Sox2 is one of them.

Overexpression of Sox2 in vitro using SC/dorsal root ganglion co-cultures had been shown to inhibit the induction of Krox-20 by cAMP and, therefore, myelination [30]. Recently, a role for Sox2 in inhibiting myelination has been shown also in vivo in mice. Sox2 was conditionally overexpressed in Schwann cells (by means of a conditionally overexpressing Sox2 allele [60], that gets activated by Cre recombination, in this case, P0-Cre [61]) and this resulted in animals smaller than their littermates, with clasping behavior (sign of abnormal motor function) and hypomyelination [31] (Figure 1D). In addition, SC overexpressing Sox2 did not associate correctly with axons and produced many processes, but little myelin compared to controls. Sox2-overexpressing SCs proliferated more, and presented features of immature non-myelinating Schwann cells, like an increase of N-cadherin (usually expressed in developing nerves and downregulated during myelination), and of Jun (a marker of promyelinating SCs) [31].

In conclusion, Sox2 appears to have a role in regulating the onset of myelination during development in the PNS.

##### Could Sox2 Have a Role in Remyelination?

Remyelination is the process by which myelinating glia of the nervous system get activated by a demyelinating lesion and stimulated to proliferate, migrate to the demyelinated area, and differentiate into new myelin-producing glia. Understanding how remyelination occurs is essential to be able to design cures for demyelinating diseases like multiple sclerosis.

Sox2 expression in OPCs and early differentiating oligodendrocytes make it a good candidate regulator of remyelination in the CNS. Indeed, in vivo studies of remyelination in mice, after induced demyelination in the CNS, have shown that Sox2 is important for OPC recruitment to the lesion site, OPC proliferation, and differentiation [24,25]. Interestingly, Sox2 is expressed in OPCs in the human brain and these OPCs can be differentiated into mature OL in vitro [62], suggesting that even in humans it could be an important regulator of remyelination.

Sox2 might also have a role in remyelination in the PNS; indeed, Sox2 overexpression in SC, following crush nerve injury, results in impairment of remyelination and of functional recovery [31]. Therefore, even in the PNS, Sox2 seems to have a role in remyelination and it appears to be required to regulate the onset of myelination [31].

### 2.2. Sox2 in Neurons

Sox2 is usually considered a marker of neural stem cells in different embryonic and adult brain regions; however, its expression has also been described in differentiated neurons in various regions of the nervous system, but until recently little was known about Sox2 function in these post mitotic neurons. Sox2 expression in pyramidal and GABAergic interneurons in the cerebral cortex and in different thalamic neurons in mice had been described [26]. In addition, a potential role of Sox2 in differentiated neurons was suggested by the identification of neurons with inclusions typical of neurodegenerative disorders and by GABAergic interneurons with aberrant morphology in the cortex of hypomorph Sox2 mutants [26,39]. Defects in post-mitotic neurons could be due to the lack of Sox2 in neural progenitors that might influence their correct differentiation, but Sox2 could also have a role after neuronal differentiation has occurred, and recently Sox2 ablation in post-mitotic neurons has been performed and this question can start to be addressed.

Three examples of the role of Sox2 in differentiated neurons will be described.

#### Thalamic Projection Neurons

Sox2 is expressed in post-mitotic neurons of thalamic sensory nuclei from late gestation to adulthood, these nuclei are the medial geniculate nucleus (MGN, that receives auditory information), the dorsolateral geniculate nucleus (dLGN, that receives visual information), and the ventro-posterior nucleus (VPN, that receives somato-sensory information from the periphery) and they project to the auditory cortex, the primary visual cortex, and the somatosensory cortex, respectively. Recently, the role of Sox2 in the neuronal development of these sensory thalamic nuclei has been investigated by means of conditional thalamic ablation within post-mitotic neurons. Particular attention has been given to the role of Sox2 in the dLGN and on how its thalamic ablation influences the development of the visual system [35].

Neurons in the dLGN receive input from the retina and project to the visual cortex. Projections from the dLGN to the visual cortex are required for the correct patterning of the visual cortex, if projections from dLGN neurons to the cortex are reduced/absent, the primary visual cortex and the adjacent higher order visual cortical areas are not correctly defined [35,63]. Sox2 deletion in these dLGN neurons, when they are already post-mitotic (by means of a thalamus specific Cre recombinase (Rorα-Cre expressed from E14.5)), results in a reduction of the size of the dLGN postnatally (Figure 2A). Interestingly, even before the reduction of the dLGN is observed, the arrival of retinal fibers to the dLGN is compromised. In addition, dLGN neuronal projections to the primary visual cortex are reduced concomitantly with the projections from the cortex to the thalamus. Even though Sox2 is not deleted in the primary visual cortex (V1), the development of V1 is strongly compromised in the Sox2 thalamic mutants, this is likely due to reduced projections from the dLGN to V1 known to be required for proper patterning of V1 [63] (Figure 2A).

Neurons in the VP also appear to require Sox2 for their correct development since their projections to the somatosensory cortex are reduced in Sox2 thalamic mutants and not correctly organized. On the other hand, projections from the MG to the auditory cortex appear less affected by Sox2 loss in the same thalamic mutants described above, but their development should be studied in more detail to properly address Sox2 requirement [35].

Retinal axons reaching the Sox2 mutant dLGN not only are reduced in number, compared to their control littermates, but they also do not segregate in the correct manner within the dLGN. In addition, a portion of axons are misrouted to the adjacent ventrolateral geniculate nucleus (vLGN). A misrouting phenotype suggests that deregulation of an axon guidance molecule could be involved. In fact, ephrin-A5 (efna5) was found to be downregulated in the Sox2 mutant dLGN, even at stages preceding the misrouting phenotype. It is thought that Efna5 could be a direct Sox2 target since Sox2 binding sites were identified in an intron, that behaves as an enhancer activated by Sox2 in in vitro co-transfection assays [18,35].

Another signaling pathway affected by thalamic Sox2 loss is the serotonin pathway. Serotonin uptake by thalamo-cortical axons in the first two postnatal weeks has been shown to be important in patterning cortical areas, in particular, the somatosensory cortex. Alterations of the serotonin pathway in mouse brains have been shown to affect the organization of barrel fields in the somatosensory cortex [64,65,66]. Serotonin uptake by dLGN neurons is affected in the thalamic Sox2 cKO, since expression of the serotonin transporters, SERT and vMAT, is strongly downregulated in the mutant dLGN. As a consequence (at least in part) of this, the amount of serotonin-positive fibers reaching the visual cortex is greatly reduced [35].

In conclusion, Sox2 expression is required in post-mitotic thalamic sensory neurons, and in particular in the visual thalamus, to properly receive and process visual information [35].

#### Clock Neurons in the Suprachiasmatic Thalamic Nucleus

Other thalamic nuclei where Sox2 expression has been described in differentiated neurons are the Suprachiasmatic nuclei (SCN)—small paired structures in the anterior hypothalamus just above the optic chiasm [67] (Figure 2B). They receive visual stimuli for light or dark through indirect or direct retina-SCN pathways. The SCN is considered the central circadian pacemaker in mammals since it is the site of a master circadian clock in the mammalian brain that generates circadian rhythms. Each neuron in the SCN is capable of independent cellular oscillations. Different clock genes have been identified that are involved in a series of transcription–translation feedback loops (TTFL) that make up the molecular clock (reviewed in reference [68]). Among the key genes involved in the molecular clock are period1 and period2.

To synchronize oscillations within the SCN, different SCN populations synthetize neuropeptides including vasoactive intestinal peptide (VIP), arginine vasopressin (AVP), and gastrin-releasing peptide (GRP). On the other hand, the production of neuropeptides such as prokineticin 2 (PK2 and Prok2) can function as outputs of the SCN to other brain regions [34,69].

Cheng et al. 2019 found that almost all PER2+ cells in the adult SCN express SOX2. To understand how Sox2 loss could affect the activity of SCN neurons, a conditional knock-out was generated in which Sox2 was deleted in all GABA-ergic interneurons (using a vescicular GABA transporter Cre recombinase). While the structural organization of the SCN was not affected by Sox2 conditional ablation, the expression of PER2 and of neuropeptide genes was greatly reduced. These mice have a deficit in light induced entrainment and display widespread changes in behavioral rhythms [34] (Figure 2B).

Sox2 was found to regulate the clock activity of SCN neurons by directly activating, in vitro and in vivo, the period2 gene. In addition, RNAseq experiments identified a reduction of transcription of other clock genes and also of neuropeptides. Therefore, Sox2 is thought to regulate signaling within the SCN nucleus and between the SCN nucleus and other parts of the brain [34].

In conclusion, Sox2 is required in clock neurons of the SCN for their proper regulation of circadian rhythms.

#### Neurons in C. elegans

The genome of the nematode *C. elegans* contains two *SoxB* genes: *sox2* (SoxB1) and *sox3* (SoxB2). On the other hand, there are 5 *SoxB* genes in mammals: *Sox1*, *Sox2*, *Sox3* (SoxB1) and *Sox14*, *Sox21* (SoxB2). *C. elegans sox2* has functional conservation with vertebrate Sox2 [36].

It has been recently shown that *C. elegans* SoxB genes are not required for neurogenesis in the developing nervous system, but for the differentiation of specific cell types. In particular, *sox2*, the SoxB1 gene, has been shown to be expressed throughout the life of several glutamatergic and cholinergic neurons and is required for their terminal differentiation. These neurons are born in sox2-null worms, but their differentiation in a specific neuron-type is compromised [37] (Figure 2C).

A role for *C. elegans sox2* in regulating the final differentiation of a specific neuronal class has also been shown for olfactory neurons. Three pairs of olfactory neurons are dedicated to sensing volatile odorants, AWA, AWB, and AWC, and their identity is regulated by regulatory programs involving neuron-type specific transcription factors. Distinct homeodomain proteins compete for cooperation with Sox2 to drive neuron specific gene expression. The Sox2/Lim4 pair drives the differentiation of the AWC neurons while the Sox2/Ceh-36 (Otx-type) pair drives the AWB differentiation program [36].

## 3. Conclusions and Perspectives

The identification of functional roles for Sox2 in specific types of differentiated neurons and glia opens a new perspective in the understanding of the function of this transcription factor in neural development and disease, enlarging Sox2 functional roles beyond those it plays within stem cells. Sox2 function in thalamic neurons is an example of an important role for Sox2 in aspects of neural cell biology central to their “differentiated” characteristics: the development of neuron-to-neuron connectivity, involving the correct establishment of the retina–thalamus–brain connections. These new roles provide an unexpected potential new explanation for the visual defects observed in Sox2 patients, acting together with Sox2 functions in the eye. The identification of the Sox2 target genes within these differentiated neural cells, and a more in-depth understanding of the gene regulatory networks mediating Sox2 function in them will provide a better understanding of Sox2 function in neural development and its pathology, with potential implications for new therapy approaches.

## Figures and Tables

**Figure 1 ijms-20-04540-f001:**
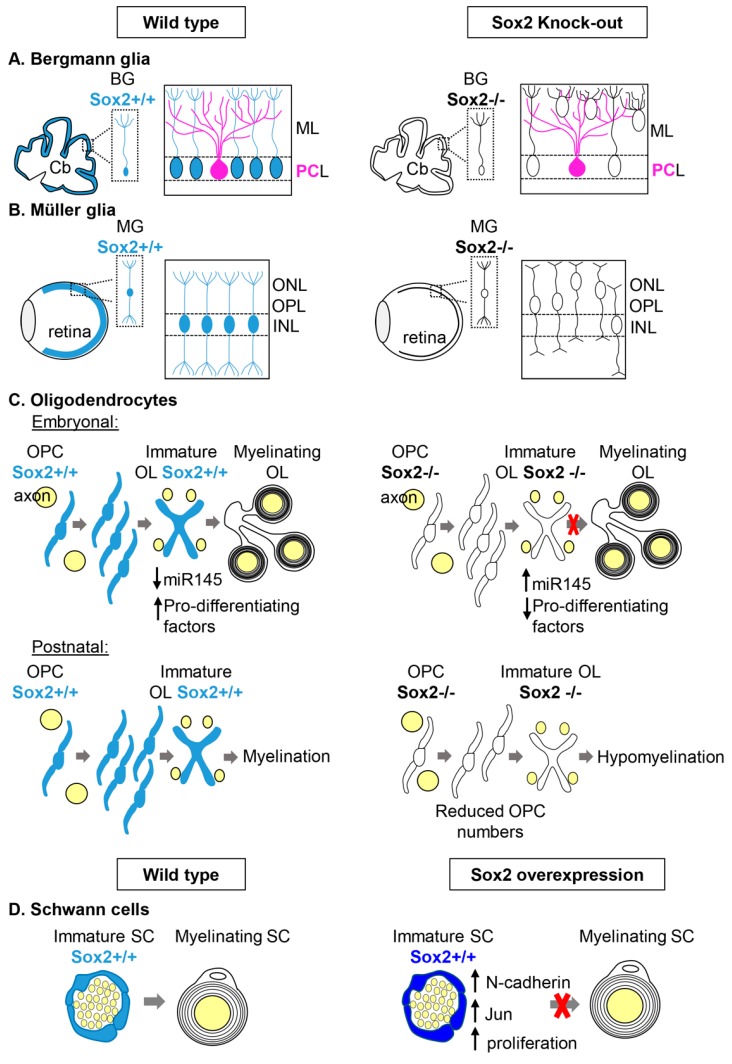
Differentiated glial cells requiring Sox2 function. Glial cells expressing Sox2 in wild type (left) are colored in blue. The corresponding cell types in Sox2 knock-out mice (right) are uncolored. (**A**) Bergmann glia in the cerebellum. Sox2-positive Bergmann glia cells (blue), forming an ordered array in wild type, that also includes Purkinje neurons (pink), are displaced and abnormal in Sox2 mutants (see text). Abbreviations: Cb, cerebellum; BG, Bergmann glia; ML: molecular layer; PCL, Purkinje cells layer. (**B**) Müller glia in the retina. Sox2-positive Müller glia cells (blue), forming an ordered array in wild type, are displaced and abnormal in Sox2 mutants, in the context of a thinner retina (see text). Abbreviations: MG, Müller glia; INL, inner nuclear layer; OPL, outer plexiform layer; ONL, outer nuclear layer. (**C**) Oligodendrocytes. In wild type embryonic oligodendrogenesis (top), Sox2-positive oligodendrocyte precursor cells and immature oligodendrocytes differentiate into oligodendrocytes, thereby, decreasing the expression of miR145, and increasing the expression of myelinating factors; in Sox2 mutants, differentiation of oligodendrocytes is abnormal, as cells retain immature features, and show higher expression of miR145 and lower expression of myelinating factors relative to the controls (see text). In postnatal oligodendrogenesis (bottom), Sox2 loss leads to reduced numbers of oligodendrocyte precursors, and to hypomyelination (see text). Abbreviations: OPC, oligodendrocyte precursor cells; OL, oligodendrocytes. (**D**) Schwann cells. Wild type immature, Sox2-positive Schwann cells produce correctly differentiated Schwann cells (left); Sox2 overexpression (right, dark blue cells) leads to cells retaining immature features, with upregulation of N-cadherin, Jun, an increase of proliferation, and failing to properly differentiate (see text). Abbreviations: SC, Schwann cells.

**Figure 2 ijms-20-04540-f002:**
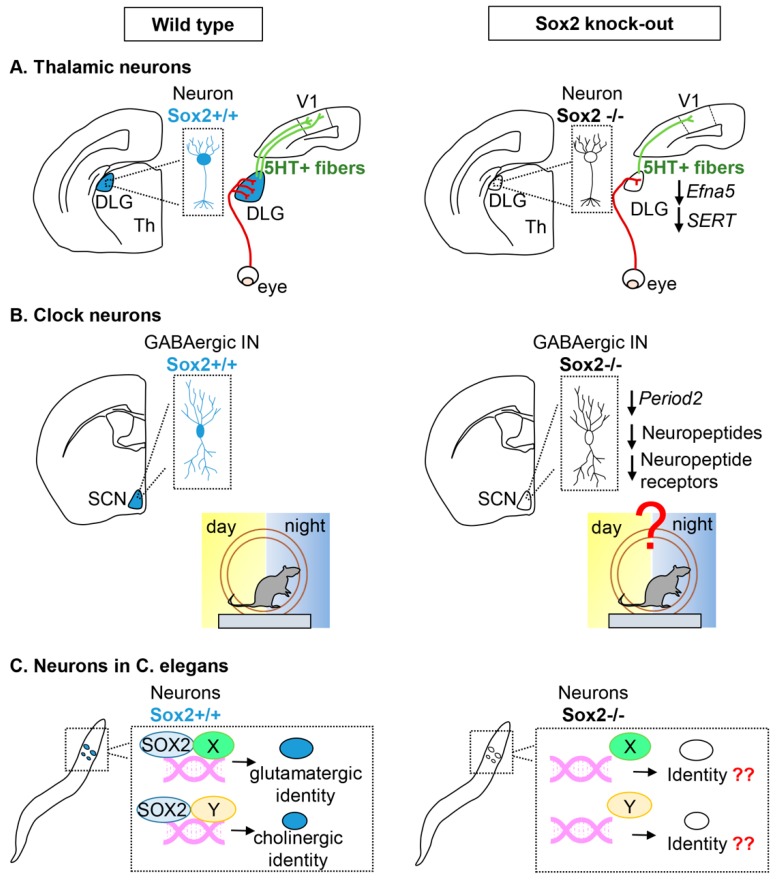
Differentiated neuronal cells requiring Sox2 function. Neuronal cells expressing Sox2 in wild type (left) are colored in blue. The corresponding cell types in Sox2 knock-out mice (right) are uncolored. (**A**) Thalamic neurons within the mouse sensory thalamic nuclei. In wild type animals, axons outgrowing from the retina reach the DLG in a spatially ordered pattern; DLG neurons develop 5HT-positive projections to the primary visual cortical area (V1). In mutant animals, the DLG is reduced, and the eye-DLG projections are reduced and disorganized; the DLG-V1 projections are also defective. A reduction of the expression of Efna5 and SERT in the mutant contributes to the defects. Abbreviations: DLG, dorsolateral geniculate nucleus; Th, thalamus; V1, primary visual area of the cerebral cortex; 5HT, 5-hydroxytryptamine, i.e., serotonin. (**B**) Clock neurons within the mouse suprachiasmatic nucleus. In wild types, Sox2-expressing GABAergic interneurons in the SCN are essential for establishing circadian rhythms. In mutants, circadian rhythms are perturbed; a decrease in the expression of Period2, and of neuropeptides and their receptors contribute to the defect. Abbreviations: SCN, suprachiasmatic nucleus; IN, interneurons. (**C**) Neurons of C. elegans. In wild types, Sox2 is expressed in specific neuronal cell types, both glutamatergic and cholinergic, and binds to DNA with different partners (X, Y). In mutants, the terminal differentiation of these neuron types is defective.

**Table 1 ijms-20-04540-t001:** SOX2 expression and function in glia and neurons.

**Glia**	**Sox2 Expression**	**Sox2 Function**	**References**
**Macroglia in the Central nervous system**			
Astrocytes	Yes	Cellular maturation and morphology (retina)	[21]
OPC and Oligodendrocytes	Yes	Proliferation, differentiation, myelination	[22,23,24,25]
Ependymal cells	Yes	Cellular structure (cilia reduction in Sox2 mutants)	[26]
Müller glia (retina)	Yes	Cellular structure and positioning	[27,28]
Bergmann glia (cerebellum)	Yes	Cellular morphology and positioning (in Sox2 mutants, defective motor control)	[29]
**Macroglia in the peripheral nervous system**			
Schwann cells	Yes	Myelination	[30,31]
Satellite cells	Yes	Survival	[32,33]
**Neurons**	**Sox2 Expression**	**Sox2 Function**	**References**
Clock neurons in Suprachiasmatic nucleus	Yes	Regulation of circadian rhythms	[34]
Sensory thalamic nuclei (including DLG)	Yes	Signalling to incoming retinal axons; DLG growth; development of thalamic neurons projections to the visual and somatosensory cortex	[35]
Specific glutamatergic and cholinergic neurons in *C. elegans*	Yes	Differentiation of specific neuron-types	[36,37]
Cholinergic amacrine neurons	Yes	Cell positioning and dendritic stratification	[38]
GABA-ergic interneurons, rare pyramidal cells in the cortex	Yes	Morphology and maturation (impaired GABAergic neuroblast migration and differentiation; intraneuronal aggregates)	[26,39]
Some striatal cells	Yes	N.A.	[26]
Thalamic cells	Yes	N.A.	[26]

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
