# Peer review of "More than just Stem Cells: Functional Roles of the Transcription Factor Sox2 in Differentiated Glia and Neurons"

_ijms, 2019, doi:10.3390/ijms20184540_

Round 1
Reviewer 1 Report
This is an excellent concise and well written review on the role of Sox2 in cells of the nervous system. The figures are also good and the only table is comprehensive. I just have a few comments.
The legend for figure 1 needs to be rewritten. Most of what is included should be in the body of the text. The legend needs to include definitions of the abbreviations used in the diagrams and references to what some of the colours represent. The legend needs to stand alone so a reader can understand the figure without constantly referring to the main text. The figure should be referred to in the text of course where appropriate. A much better job is done with the legend for figure 2.
I would suggest using the term 'human' rather than 'man', for example on line 18 in the abstract and lines 43 and 58.
On line 109 should Sox be Sox2?
I'm not sure why large spaces are used at times, for example on lines 97-99, and line 321, is this necessary?
Author Response
Dear Reviewer 1,
Thank you for your comments.
We made the following changes as you suggested:
The legend to Figure 1 in the body of the manuscript will be substituted with legend to Figure 1 found at the end of the manuscript. There was a formatting error so part of the text of the manuscript was converted into the legend to Figure 1. Thanks for pointing this out. We substituted “man” with “human” on line 18, 43 and 58. We changed “Sox” to “Sox2” on line 109. We removed extra spaces throughout the manuscript including the lines you highlighted.
We made the following changes that we think will improve the manuscript:
We substituted Figure 1 and Figure 2 with new figures that contain the same drawings, but are more organized and we think would be more clear for the reader. We inserted them at the end of the manuscript after the legends. We corrected spelling and punctuation mistakes. We added one reference to Table 1 and updated the reference list at the end of the manuscript in order to include all the references from Table 1. We added numbers to the different sections of the manuscript: 1. Introduction, 2. Sox2 function in differentiated glia and neurons, 2.1 Sox2 in glial cells, 2.2 Sox2 in neurons, 3. Conclusions and perspectives.
Sincerely,
Sara Mercurio
Reviewer 2 Report
The review article submitted by Sara Mercurio et al is a comprehensive presentation on transcription factor Sox2 and authors have compiled the studies on function aspect of transcription factor Sox2 in glia and neurons.
The overalll report has been presented in well organized manner and provide analysis of so far published relevenat studies.
I would recommend this manuscript to be accepted for publication.
Author Response
Dear Reviewer 2,
Thank you for your positive comments on our manuscript.
We made the following changes that we think will improve the manuscript:
We substituted Figure 1 and Figure 2 with new figures that contain the same drawings, but are more organized and we think would be more clear for the reader. We inserted them at the end of the manuscript after the legends. We corrected spelling and punctuation mistakes. We added one reference to Table 1 and updated the reference list at the end of the manuscript in order to include all the references from Table 1. We added numbers to the different sections of the manuscript: 1. Introduction, 2. Sox2 function in differentiated glia and neurons, 2.1 Sox2 in glial cells, 2.2 Sox2 in neurons, 3. Conclusions and perspectives.
Sincerely,
Sara Mercurio